# Single prolonged stress blocks sleep homeostasis and pre-trauma sleep deprivation does not exacerbate the severity of trauma-induced fear-associated memory impairments

Christopher J. Davis, Jason R. Gerstner, William M. Vanderheyden *

Department of Biomedical Sciences, WSU Health Sciences Spokane, Elson S. Floyd College of Medicine, Spokane, Washington, United States of America

* w.vanderheyden@wsu.edu

**Data Availability Statement:** All relevant data are within the manuscript and its Supporting Information files.

## Abstract

Sleep is intimately linked to cognitive performance and exposure to traumatic stress that leads to post-traumatic stress disorder (PTSD) impairs both sleep and cognitive function. However, the contribution of pre-trauma sleep loss to subsequent trauma-dependent fear-associated memory impairment remains unstudied. We hypothesized that sleep deprivation (SD) prior to trauma exposure may increase the severity of a PTSD-like phenotype in rats exposed to single prolonged stress (SPS), a rodent model of PTSD. Rats were exposed to SPS alone, SD alone, or a combination of SPS+SD and measures of fear-associated memory impairments and vigilance state changes were compared to a group of control animals not exposed to SPS or SD. We found that SPS, and SPS+SD animals showed impaired fear-associated memory processing and that the addition of SD to SPS did not further exaggerate the effect of SPS alone. Additionally, the combination of SPS with SD results in a unique homeostatic sleep duration phenotype when compared to SD, SPS, or control animals. SPS exposure following SD represses homeostatic rebound and eliminates sleep-deprivation-induced increases in NREM sleep delta power. This work identifies a unique time frame where trauma exposure and sleep interact and identifies this window of time as a potential therapeutic treatment window for staving off the negative consequences of trauma exposure.

## Introduction

According to the National Institutes of Health and Institute of Medicine, 50–70 million US adults suffer from a sleep disorder that effectively interferes with getting sufficient nighttime sleep. Lack of sleep impairs physical and cognitive functioning and sleep disturbances are often comorbid with and may exacerbate other mental or physical health impairments [1–4]. However, it is unknown if sleep disturbances contribute to increased susceptibility to psychiatric illness. For example, it is unclear if poor sleep prior to trauma exposure increases the

**Funding:** This work was supported by the Department of Defense, Congressionally Directed Medical Research Program, Discovery Award # W81XWH-18-1-0378. The funders had no role in study design, data collection and analysis, decision to publish, or preparation of the manuscript.

**Competing interests:** The authors have declared that no competing interests exist.

likelihood of developing post-traumatic stress disorder (PTSD). PTSD occurs as the result of experiencing a physical or psychological trauma [5] and presents with hallmark traits of sleep disturbances including insomnia, nightmares, and sleep fragmentation [6–8]. Sleep disturbances are highly prevalent in PTSD, in fact, as many as 70–91% of PTSD patients have reported difficulty falling asleep and reduced sleep efficiency [6]. Poor sleep has been hypothesized to exaggerate PTSD symptoms creating a reciprocal relationship between sleep and PTSD wherein trauma exposure impairs sleep and trauma-induced sleep disorders increase the severity of PTSD [6, 9]. Still, the contribution of sleep loss prior to trauma exposure on the development or susceptibility to PTSD is unknown. Therefore, we examined whether sleep deprivation prior to trauma exposure alters subsequent sleep or the development of trauma-induced fear-associated memory impairments.

To achieve our goals, we utilized the single prolonged stress (SPS) rat model of PTSD. SPS is a combination of stressors that includes physical restraint, 20 min of forced swim, exposure to ether vapors until anesthetized, and 7 days of social isolation [10], and has consistently shown good face validity to the human condition of PTSD [10–13]. In this model, fear-associated memory impairments are used as a surrogate for the severity of the PTSD-like phenotype [14, 15] giving us a reliable metric to assess the contribution of pre-trauma sleep loss to the development of a PTSD-like phenotype.

In the present study, we compared measures on a fear-associated memory task in SPS exposed animals, sleep deprived (SD) animals, and SPS+SD treated groups to control animals. Sleep deprivation occurred during the animal's primary sleep phase (ZT0-12, lights on) and because SPS exposure occurred at ZT0 in our previously published work and due to circadian effects on learning and memory [16, 17], we also hypothesized that SPS exposure at ZT12, the onset of the predominantly active period in nocturnal rodents may have unique contributions to the effect of SPS trauma exposure on fear-associated memory.

## Materials and methods

### Animals

All animal procedures were carried out in accordance with the National Institutes of Health *Guide for the Care and Use of Laboratory Animals* and approved by the *Washington State University Committee on the Use and Care of Laboratory Animals.*

Male, Long Evans rats were used for all experiments to eliminate the known impact of estrous cycle hormones on sleep and behavior [18–20]. Rats (60–90 days old) were housed in temperature (21˚C -24˚C) and humidity-controlled (30–70%) rooms on a 12:12 light-dark cycle and given *ad-libitum* access to food and water. The animals underwent one survival surgery to implant electroencephalographic (EEG) and electromyographic (EMG) recording electrodes (described below). Animals were housed singly following surgical procedures.

### Experimental timeline

Animals were randomly assigned to the following 4 groups: control (no SPS, no SD, n = 9), SPS (SPS only, n = 9), SD (SD only, n = 9) or a combination of SPS and SD (SPS+SD, n = 14). All animals were recorded for a 24h baseline. Control animals were left untouched for the remainder of the experiment whereas, the sleep deprived animals (SD, and SPS+SD) experienced 12h of sleep loss during their primary sleep phase (ZT0-12, labeled SD in Fig 1). This was followed by SPS trauma exposure occurred at the transition from lights-on to lights-off (ZT12) in the SPS and SPS+SD groups (blue box in Fig 1 represents the time point of SPS treatment). After SPS, all groups were isolated for 7 more days. On the eighth day, all groups

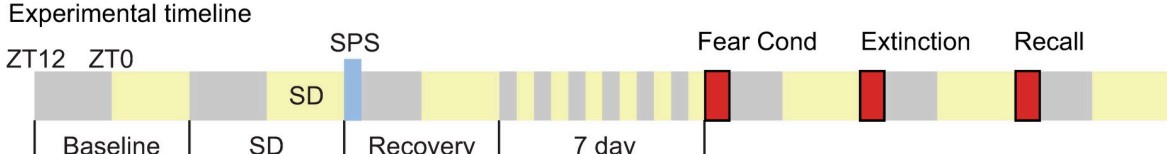

**Fig 1. Schematic representation of experimental timeline.** Baseline sleep was recorded for 24h. On the following day, SD animals underwent 12h of sleep deprivation from ZT0-ZT12 during their primary sleep phase while non-deprived animals were left alone. At the start of the 3rd day of recording, the SPS and SPS+SD groups were exposed to SPS at ZT12 (blue box) and then returned to their home cage for 7 subsequent days. At the conclusion of the 7-day SPS incubation period, all four groups of animals were exposed to the fear conditioning/extinction/recall paradigm (red boxes) as described in the methods section.

underwent fear-associated memory testing which included one session each of fear conditioning, extinction and recall as described below (red boxes, Fig 1).

## Electrophysiology

Survival stereotaxic surgery was performed to implant the EEG/EMG sleep recording electrodes as previously published [13, 21]. Aseptic surgeries were performed under isoflurane anesthesia. A midsagittal incision was made on the top of the skull and the skin was retracted. After cleaning the surface of the skull, 4 holes were drilled through the cranium and screw electrodes (Plastics One, Roanoke, VA) were inserted bilaterally over the frontal area (±2.5 mm lateral to midline, 2.5 mm anterior to Bregma) and over the hippocampal area (±2.5 mm lateral to midline, 3.5 mm posterior to Bregma) for EEG recordings. Two flexible wire electrodes were threaded through the dorsal neck muscles for EMG recordings. Gold pins were connected to the ends of each electrode then placed into a six-pin connector (Plastics One, Roanoke, VA) which was attached to the skull via light-curable dental acrylic. Electronic connections were finalized through the six-pin connector to a Tucker Davis Technology (TDT) (Alachua, FL) electrophysiology recording device. Rats were given at least 10 days to recover from surgery prior to beginning the experiment. Animal well-being was assessed daily during the surgical recovery period. Any sign of illness or pain, including decreased motility and responsiveness, vocalizations, lack of appetite, decreased grooming, etc. was noted and treated in consultation with veterinary staff.

## Sleep recording and analysis

Following recovery from surgery, animals were housed individually and connected to the TDT recording system via a lightweight, flexible tether attached to a commutator (Sparkfun.com, Slip Ring) for free movement within the cage. The recording system was used to sample signals at 333 Hz, filtered between 0.1–100 Hz and amplified. Prior to analysis, signals were down sampled to 250 Hz. The two EEG electrodes were differentially referenced to obtain one EEG channel. Two EMG channels were also differentially referenced to obtain the EMG signal. Animals were given 48h to acclimate to the tethers prior to beginning baseline recordings. During the acclimation period, animals were supplied food to last the duration of the EEG/EMG experimental recordings. While connected to tethers, animals were monitored daily for food, water, and health via visual inspection by a video monitoring system to avoid disturbing the animals.

Collected data were transferred from the recording PC, stored onto disk, and scored off-line in 10-second epochs to determine sleep/waking state using Sleep Sign software (Nagano, Japan). Three vigilance states were assigned: Wake, REM sleep and NREM sleep. Wake consists of visible EEG theta activity and high EMG activity, REM sleep consists of clear, sustained

EEG theta activity and phasic muscle twitches on a background of low EMG, NREM sleep consists of high amplitude, synchronized EEG and low EMG activity.

EEG and EMG signals were recorded for 24h of baseline and during the SD, after which the animals were unhooked from the recording system and single prolonged stress (SPS, described below) was performed. Following SPS, animals were reconnected to the recording system and seven subsequent days were recorded and scored. Data collected after SPS were compared to the baseline recording day using Graphpad Prism software by two-way repeated measures ANOVA followed by Bonferroni post hoc comparisons of each day to the baseline. Sleep states were quantified as an average duration spent in state per hour (in seconds) over the light phase (ZT0-12) and dark phase (ZT12-24).

EEG spectral analysis was performed using SleepSign Software and averaged in 12- and 24h artifact-free epochs across the experimental condition. Spectral power was examined by individual sleep state in half Hz bins in the 0–20 Hz range using a Hanning window filter. We also report the average spectral power of theta (5-8Hz) during REM sleep and delta (1-4Hz) during NREM sleep. Student's t-tests were performed on these average data between baseline and recovery days as indicated.

## Sleep deprivation

Animals were sleep deprived for 12h (ZT0-12) by gentle handling as previously published [22]. Animals were stimulated by gently touching the tail, body, or nose of the animal with a wooden tongue depressor or small paintbrush when the animal appeared to fall asleep or when EEG/EMG indicated a transition to a sleep state. Instrumented animals remained connected to the EEG/EMG recording device during this manipulation.

## Single prolonged stress

Single prolonged stress was performed as previously published [10, 13, 21]. Briefly, animals were exposed to 3 successive stressors at the start of the dark phase. First, physical restraint was performed for 2h in custom built plexi-glass restraining devices. Next, the animals were placed in a (63.5W x 43D x 40H cm) plastic bin containing 21–24˚C water and were forced to swim in groups of 6–8 for 20 min. Following a 15-min recuperation period in a towel-lined bin, the animals were exposed to 60ml of ether vapors in a 2000cc isolation chamber until fully anesthetized (<5 min). Ether exposure is a critical component in the development of the PTSD phenotype in rats. Substitution of an alternative anesthetic such as isoflurane for ether is insufficient to cause extinction retention deficits in fear-associated memory processing [14]. After which the animals were returned to their EEG/EMG recording-cage where they were isolated for the following seven days (as shown in Fig 1).

## Fear conditioning, fear extinction, and extinction recall

At the conclusion of the EEG/EMG recording, fear-associated memory tests were conducted. Fear conditioning, extinction, and extinction recall, hereafter referred to as recall, were performed as previously published [13, 15, 21]. All fear conditioning, extinction, and recall experiments were performed in four identical Noldus fear conditioning chambers (Wageningen, the Netherlands) (30.5W x 25.5D x 30.5H cm) containing a Shock Floor with current carrying metal bars, a wall-mounted speaker and in-chamber UV and white lighting. Test cages were housed in sound-attenuating boxes. Tones (2000 Hz, 80 dB) were delivered via speakers mounted in the housing of the test cages and controlled by data acquisition software (Noldus Ethovision XT14). Ceiling mounted cameras recorded behavior for analysis and Noldus Ethovision software was used to quantify freezing levels. Freezing values were analyzed with

Graphpad Prism software by one-way and two-way ANOVA where appropriate, followed by Bonferroni post-hoc comparisons.

Two unique contexts were created by manipulating olfactory and visual cues. Context A consisted of 50 ml of 1% acetic acid solution placed in a small dish next to the test cage using standard lighting conditions of the above-mentioned housing boxes. Context B consisted of 50 ml a 1% ammonium hydroxide solution placed in a small dish above the test cage along with a checkerboard patterned paper placed on the chamber walls to alter the visual context.

Fear conditioned animals were exposed to five, 1 mA, 1s foot-shocks paired with the cessation of a 10s80 dB tone in Context B. The first tone was presented 180safter the animal was placed in the test cage and the subsequent tones occurred with a 60sinter-tone interval. For all phases, baseline freezing was assessed for 180sprior to the presentation of any tones, and the inter-tone interval was 60s. 60s after the last tone, animals were removed to their home cages. Fear extinction was conducted 24h after fear conditioning and was performed in a distinctly different context (Context A). Fear extinction consisted of presentation of 30 tones (60sinter-tone interval), without the paired foot-shock. Recall was assessed 24h after extinction and consisted of the animals being placed back into the fear extinction context (Context A) for 10 tones (60sinter-tone interval), again without foot shock.

## Results

The average percent time freezing during the fear conditioning training was significantly increased in SPS (72.99 ± 9.5%) and SPS+SD (64.6 ± 11.3%) exposed animals compared to controls (45.67 ± 8.0%). SD animals (52.4 ± 9.3%) did not differ from controls in response to fear conditioning (Fig 2A). During the extinction phase, the average percent time freezing was higher in the SPS exposed (60.7 ± 3.2%), SD (48.2 ± 2.1%) and SPS+SD (58.5 ± 3.6%) animals compared to controls (40.3 ± 3.0%) (Fig 2B). Finally, during the fear recall phase, the average percent time freezing was higher in the SPS exposed (49.7 ± 4.6%) and the SPS+SD (41.6 ± 4.1%) groups compared to controls (27.4 ± 2.4%). SD animals (26.1 ± 4.1%) did not differ from controls in response to fear recall (Fig 2C).

To further examine the time course of fear associated memory impairments, we examined group by time interactions on fear conditioning, extinction and recall in these 4 groups of animals. We found a significant group by time interaction on each of these days (Fig 3A–3C). The largest effect was seen on the fear conditioning day where, compared to control animals, the SPS exposed animals showed enhanced freezing at each of the 5 time points measured (* = p < 0.05, Fig 3A). The SPS+SD animals showed enhanced freezing at the 3–5 time points compared to controls (+ = p < 0.05). The SD group showed a significant increase in freezing at the 5[th] time point of fear conditioning (# = p < 0.05). Although significant group x time interactions were detected on the extinction and recall day, Bonferroni post hoc analysis revealed that no individual time points reached statistical significance compared to control animals (Fig 3B and 3C).

Next, we examined the change in sleep duration in response to SPS, SD, and SPS+SD. Physiological EEG/EMG recordings were made over a baseline day, the sleep deprivation day, and finally on the trauma exposure/sleep recovery day (Fig 4A). SD and SPS+SD groups showed very little, if any, REM sleep on the deprivation day. In the 12h following sleep deprivation, the SD group showed a >250% increase in REM sleep duration compared to baseline. Interestingly, the SPS+SD group only showed a 150% increase in REM sleep duration suggesting that SPS suppresses normal REM sleep rebound. Additionally, REM sleep rebound in the SPS+SD group persisted in the subsequent 12h period afterwards. SPS alone was insufficient to cause a change in REM sleep duration immediately following trauma but resulted in increased REM

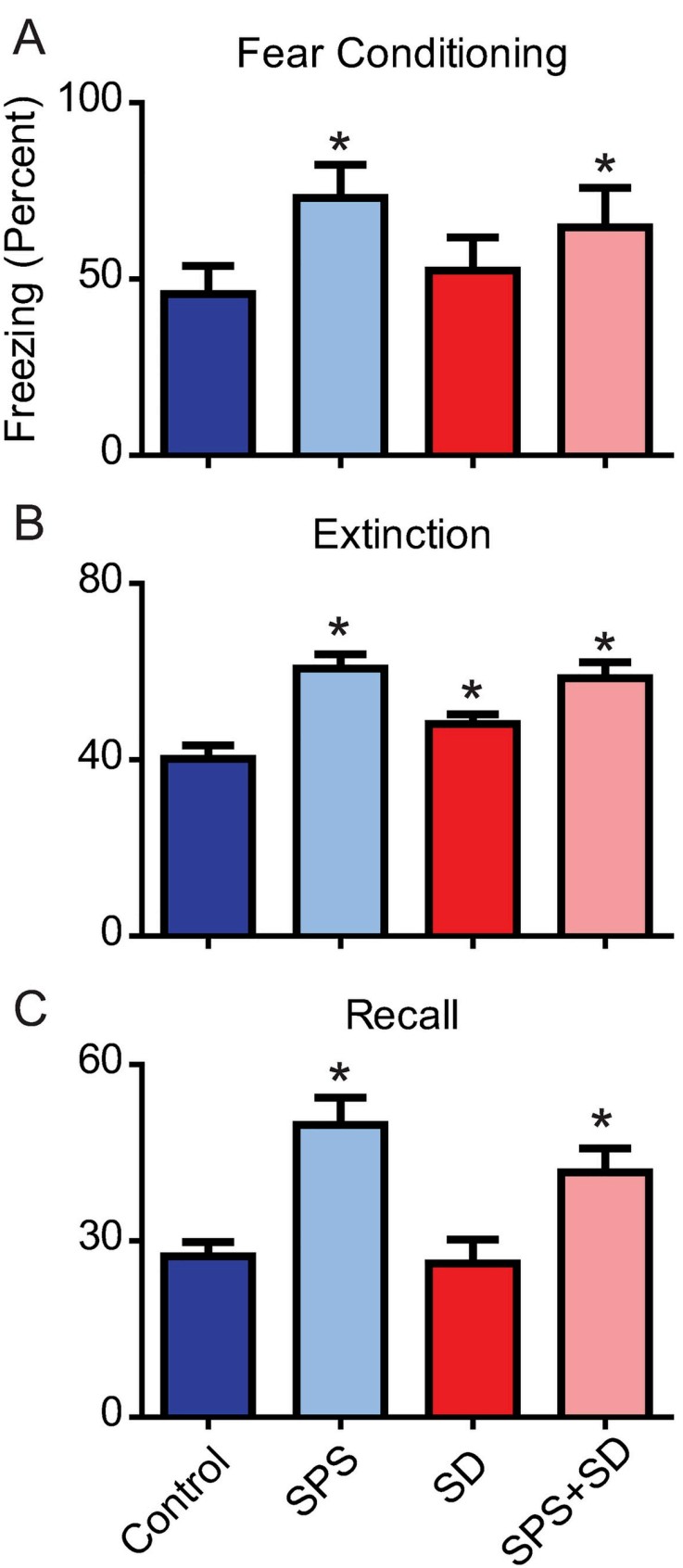

**Fig 2. SPS impairs fear-associated memory and sleep deprivation does not further exaggerate fear-associated memory impairments.** Each bar represents the average freezing duration (percent) over the fear conditioning (A), extinction (B), and recall (C) phases of the fear-associated memory test. Control animals (dark blue bars) were compared to animals exposed to SPS (light blue bars), SD (dark red bars), or a combination of SPS+SD (light red bars). A one-way ANOVA was used to detect significant effects of treatment, followed by Bonferroni post-hoc analysis., A) A significant main effect of treatment ($F_{(3,19)} = 12.95$, $P = 0.0005$) was found during the fear conditioning phase. Post-hoc analysis revealed that SPS (light blue) and SPS+SD (light red) animals froze significantly more than control animals (dark blue). B) A significant main effect of treatment was found during the extinction phase ($F_{(3,119)} = 35.42$, $P < 0.0001$). Bonferroni post-hoc analysis revealed that the SPS exposed, SD, and SPS+SD groups froze significantly more than control animals. C) A significant main effect of treatment was found during the recall phase ($F_{(3,39)} = 26.28$, $P < 0.0001$). Bonferroni post-hoc analysis showed that SPS treated (light blue) and SPS+SD (light red) treated animals froze significantly more than control animals (dark blue) * = $P < 0.05$.

sleep duration 12h later (Fig 4B). These data are in agreement with previously published work that shows a 12h delay in increased REM sleep duration in response to trauma exposure [13, 21].

As with REM sleep, NREM sleep duration was also effectively reduced to zero in response to sleep deprivation in the SD and SPS+SD groups on the deprivation day. During the first 12h of the recovery sleep phase, NREM sleep rebounded about 150% in control animals but was completely blocked when SPS was presented at the conclusion of the sleep deprivation (Fig 4C).

Sleep deprivation in the SD and SPS+SD groups resulted in increased waking (Fig 4D). In the subsequent 12h recovery time, the SD group showed a significant reduction in Wake duration, coinciding with increased sleep rebound. This reduction in Wake was not seen when SPS was performed at the conclusion of the sleep deprivation.

We examined EEG spectral power over the three days of the experiment. Control animals (Fig 5A and 5B) did not show any difference in NREM or REM spectral power over the 3 days. We found that SPS alone had no effect on NREM spectral power (Fig 5C) but resulted in reduced REM theta power in the 12h immediately after the trauma (Fig 5D, inset). In contrast, sleep deprivation alone was sufficient to increase NREM delta power (Fig 5E) and the combination of SPS+SD effectively eliminated the effects of each of these individual treatments (Fig 5G and 5H).

## Discussion

The goal of this work was to examine if sleep deprivation prior to trauma exposure leads to increased severity of PTSD-like phenotype in SPS exposed rats. In previous studies, increased freezing on the recall day has served as an index of PTSD severity [13–15]. SPS typically results in animals with more severe extinction retention deficits that is manifest as increased freezing on the recall day. However, the data presented here show that sleep loss prior to trauma exposure did not further exaggerate the fear-associated memory impairments found by exposure to the trauma alone as there is no difference in freezing between the SPS and the SPS+SD group on the fear recall day.

However, the SPS exposed and SPS+SD animals showed significantly more freezing on the fear conditioning day and the extinction day compared to control and SD groups. This result was novel and unique to this set of experiments as previous groups showed no differences in acquisition or extinction of fear memory in SPS treated rats [13–15].Given that these studies were performed precisely as previously published, with the exception of the time of day that they were performed, leads to the suggestion that the timing of trauma exposure affects fear-associated memory acquisition and extinction processing in this model. To our knowledge, the effect of trauma exposure at different times of the day has not been systematically

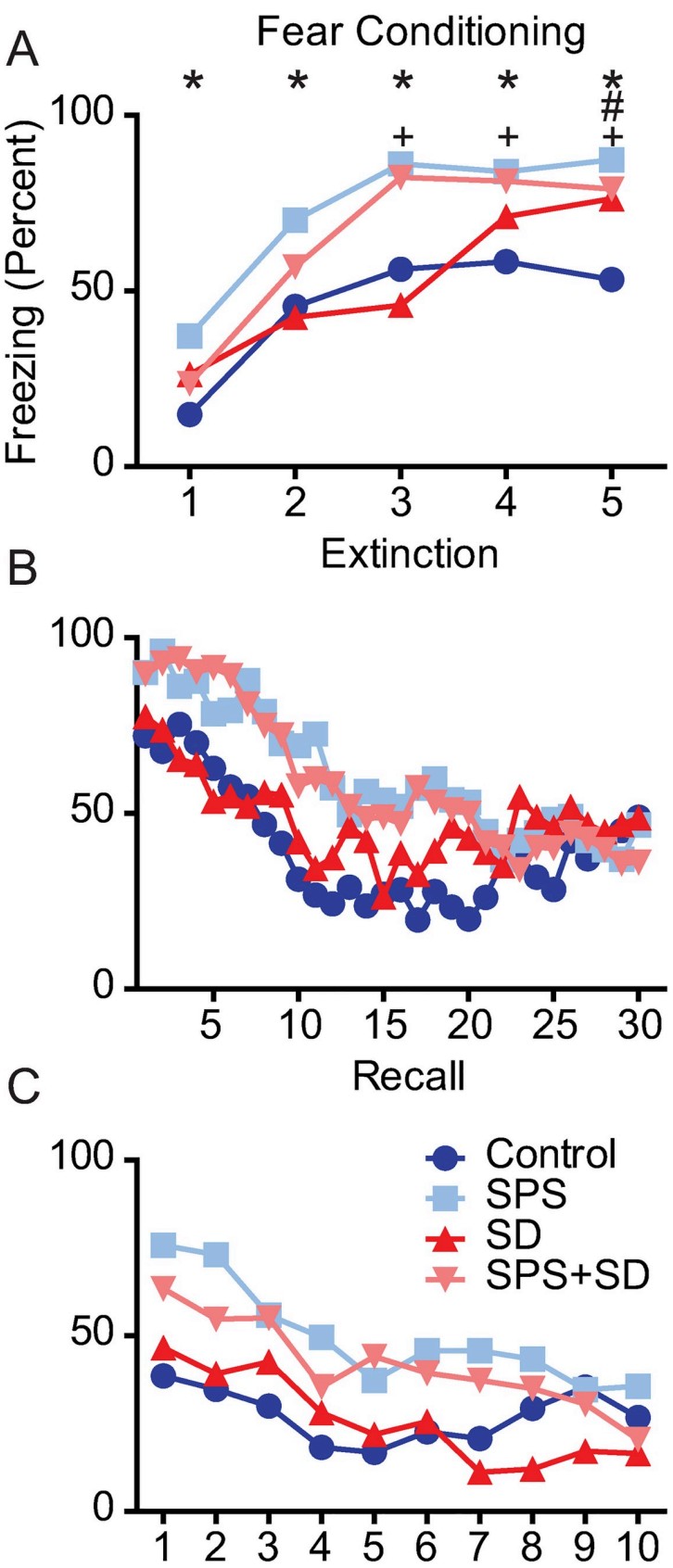

**Fig 3. SPS exposure significantly alters the timeline of fear-associated memory training, extinction, and recall.** Each line represents the freezing duration (percent) over each 1 min period of the fear conditioning, extinction, and recall phase of the fear-associated memory test. Each phase was analyzed with two-way ANOVA followed by Bonferroni post-hoc analysis that compared control animals (dark blue circles) to animals exposed to SPS (light blue squares), or SD (dark red triangles), or a combination of SPS+SD (light red triangles). A) two-way ANOVA revealed a significant group x time interaction ($F_{(12,148)} = 3.14$, $P = 0.0005$) on the fear conditioning day, B) the extinction day, ($F_{(187,1073)} = 1.54$, $P = 0.0015$) and, C) the recall day ($F_{(27,333)} = 1.63$, $P = 0.027$). Bonferroni post-hoc comparisons revealed significant differences from (dark blue) control animals ($P < 0.05$) and are denoted by * for SPS (light blue), # for SD (red), and + for SPS+SD (light red) animals.

investigated, while much effort has gone into examining the contribution of the circadian system on other learning and memory processes [16, 23–26]. Therefore, the primary findings of our study are two-fold, 1) we provide evidence that sleep loss prior to trauma exposure does not further exaggerate SPS-induced fear extinction retention deficits, and 2) we show that trauma exposure at ZT12 results in heightened SPS-induced fear-conditioning and fear-extinction acquisition freezing rates.

Using cued fear conditioning, we determine PTSD severity by comparing freezing on the fear recall day of the fear-associated memory task in SPS exposed and control rats. Fear conditioning, pairing an audible tone with a foot shock, typically results in increased freezing behavior that does not differ between SPS and control animals [13–15]. However, in these experiments, SPS treated animals and SPS+SD animals both show increased freezing in response to fear conditioning compared to control (and SD alone) animals (Fig 2A). These data indicate that trauma induced fear-associated memory processing may function differently at ZT12, than at ZT0. Identifying a mechanism or understanding this unique phenotype is an area of active research by our group and further examination is required to identify the brain regions and molecules by which this phenotype develops.

In the absence of SPS, sleep deprivation resulted in a robust REM and NREM homeostatic rebound. Interestingly, when SPS was added at the conclusion of the sleep deprivation (SPS +SD) REM sleep rebound was significantly attenuated (Fig 4B) and NREM sleep rebound was completely blocked. Remarkably, REM sleep was also increased in the subsequent 12h period after trauma exposure in both the SPS group and the combined SPS+SD group. The differences in sleep profile that emerges following these treatments may serve to identify critical windows-in-time for the application of sleep-specific interventions to stave off the negative consequences of trauma exposure. For instance, it may be possible to manipulate sleep time via optogenetics [27] or pharmacology during these critical time periods in order to help to assign function to each of these physiological sleep properties.

In addition to measuring sleep duration, we also examined changes in delta and theta spectral power during NREM and REM respectively. SPS reduces REM sleep theta band power while sleep deprivation increases NREM delta power in the 12h immediately after these manipulations. However, these spectral power phenotypes were abolished when SPS and SD were combined (SPS+SD). SPS is unique from other forms of stress and our application of it has identified a unique sleep signature that may be important for the development of PTSD.

Previous work has shown that stressful stimuli can change sleep [28–32]. This work has used a diverse collection of "stressors" (e.g., foot shock, predator scent, physical restraint) to change sleep. However, SPS is unique from these stressors in that it induces a PTSD-like phenotype [10] and creates long term changes in sleep architecture [13] and is different than an acute response to stress that does not induce a PTSD-like phenotype. For example, previous work showed social conflict stress, a stressor not known to induce PTSD, resulted in increased slow wave activity during NREM sleep [33]. Interestingly, when sleep deprivation was added

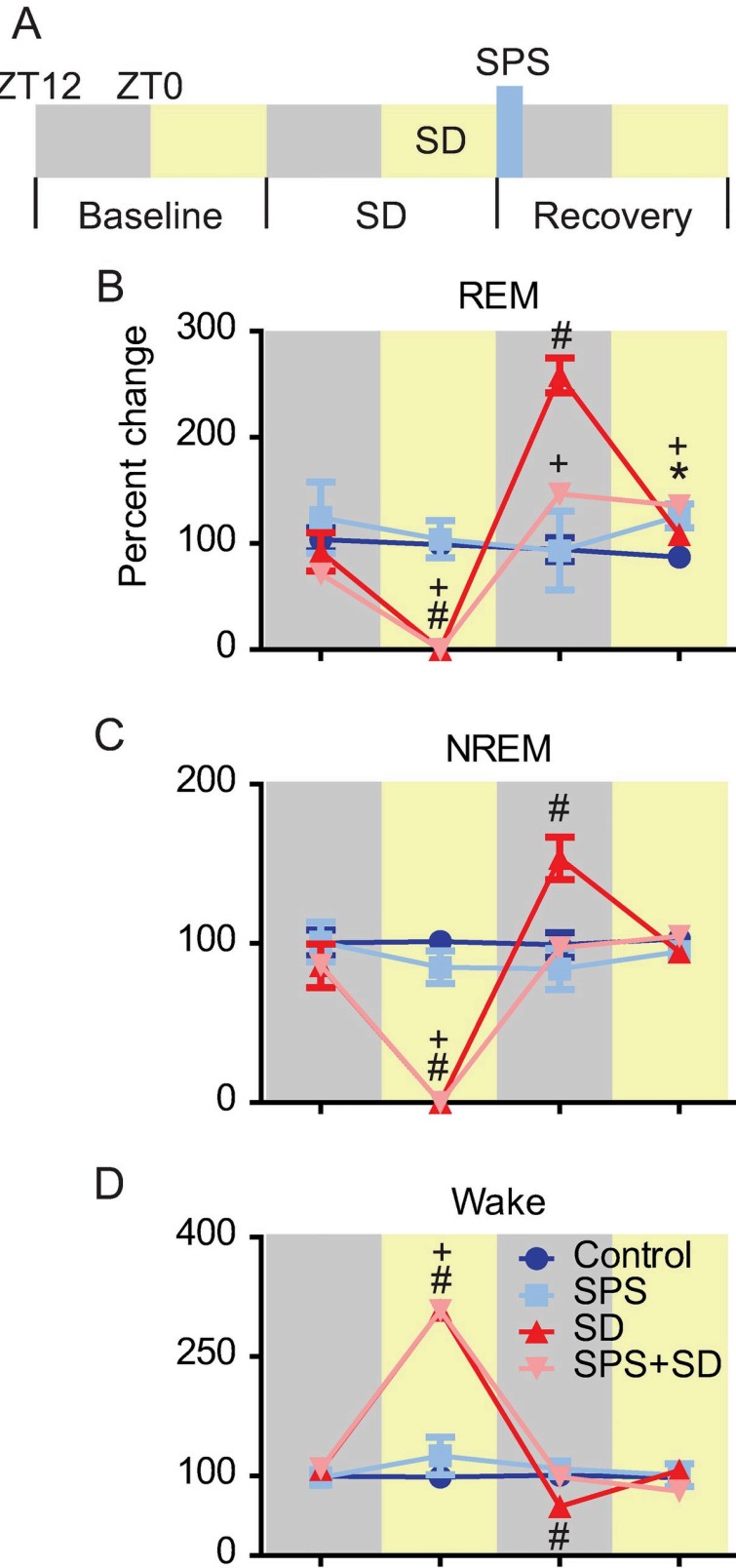

**Fig 4. SPS exposure alters sleep-deprivation-induced rebound.** A) Experimental timeline. 24h of (baseline) sleep was recorded for all groups of animals. On the following day, sleep deprived animals experienced 12h of sleep loss during the light phase (SD). On the final day of recording (recovery) SPS was performed at the transition from lights on to

lights off (blue bar). Each line represents the percent change in REM, NREM, and Wake time compared to baseline over the sleep deprivation and recovery day in control (blue), SPS (light blue), SD (red), and SPS+SD (light red) exposed animals. Each phase was analyzed with two-way ANOVA followed by Bonferroni post-hoc analysis. Significant differences from (dark blue) control animals (P <0.05) are denoted by * for SPS, # for SD, and + for SPS +SD animals. B) REM sleep showed a significant group x time interaction (F(9,51) = 12.24, P < 0.0001). Bonferroni post-hoc analysis identified significant reductions in REM sleep on the SD day for the SD (red line) and SPS+SD. SD and SPS+SD animals showed a significant increase in REM sleep during the dark phase immediately after the sleep deprivation. The SD+SPS and SPS exposed groups showed increased REM sleep during the light phase after SPS. C) NREM sleep showed a significant group x time interaction (F(9,51) = 19.99, P < 0.0001). Bonferroni post-hoc analysis identified significant reductions in NREM sleep on the SD day for the SD and SPS+SD. SD animals showed a significant increase in NREM sleep during the dark phase immediately after the sleep loss. D) Wake duration showed a significant group x time interaction (F(9,51) = 47.03, P < 0.0001). Bonferroni post-hoc analysis identified significant increase in wake duration on the SD day for the SD and SPS+SD animals. SD animals showed a significant decrease in wake duration during the dark phase immediately after the sleep loss.

to this social conflict stressor, slow wave activity was further increased above that of sleep deprived animals alone [34]. In our studies here, we find that NREM sleep rebound is completely abolished when SPS is added after sleep deprivation. That NREM sleep may be critical to recovery from trauma exposure and further work is required to examine the contribution of this NREM sleep on the development of PTSD. Therefore, our use of SPS to examine the contribution of sleep deprivation prior to trauma exposure on subsequent sleep time and sleep EEG spectral power is novel and may have better translational properties to the human condition of PTSD.

We live in a world where sleep deprivation is increasingly common. The Centers for Disease Control and Prevention estimates that up to a third of US adults get less than the required amount of sleep (www.cdc.gov/sleep). This sleep loss has severe consequences on emotional [35] and cognitive function [36, 37] and has resulted in a societal need to appreciate and respect sleep as vital to our functioning [38]. Sleep may be particularly relevant to trauma exposed populations who experience significant trauma-induced sleep disturbances [6, 9, 39]. This is especially true given that fixing trauma-induced sleep disturbances seems to assist in preserving function as our group has recently shown that optogenetically enhancing sleep in rats after SPS trauma exposure can significantly reduce trauma-induced fear-associated memory impairments [21]. Here we have used acute sleep deprivation to assess the impact of extended wakefulness near the time of trauma exposure on cognitive function. In the present study we find that acute sleep loss is ineffective to enhance fear-associated memory impairments. Future studies should compare the effects between circadian disruption and varying degrees of sleep deprivation pre- and post- trauma, as well as the consequences of mimicking sleep fragmentation and prolonged sleep restriction observed clinically by trauma-exposed individuals.

Understanding the role of sleep in PTSD is particularly relevant for military personnel, whose lifetime prevalence rates of PTSD range from 10–30% depending on the era of military deployment and whose sleep may suffer both while in combat zones and after tours of duty (www.ptsd.va.gov). In addition, 1–10% of non-military U.S. adults are also affected by PTSD and these cumulative populations account for an estimated $42 billion dollars per year in medical and prescription drug costs, lost wages, and mortality costs. PTSD and the associated sleep disturbances are a costly and immediate public health problem. This work has been foundational in improving our understanding of the interaction of trauma and sleep, however, further studies are required to aid in the development of sleep therapeutics, to identify sleep-related biomarkers to reduce PTSD prevalence, or to help currently affected individuals regain functioning.

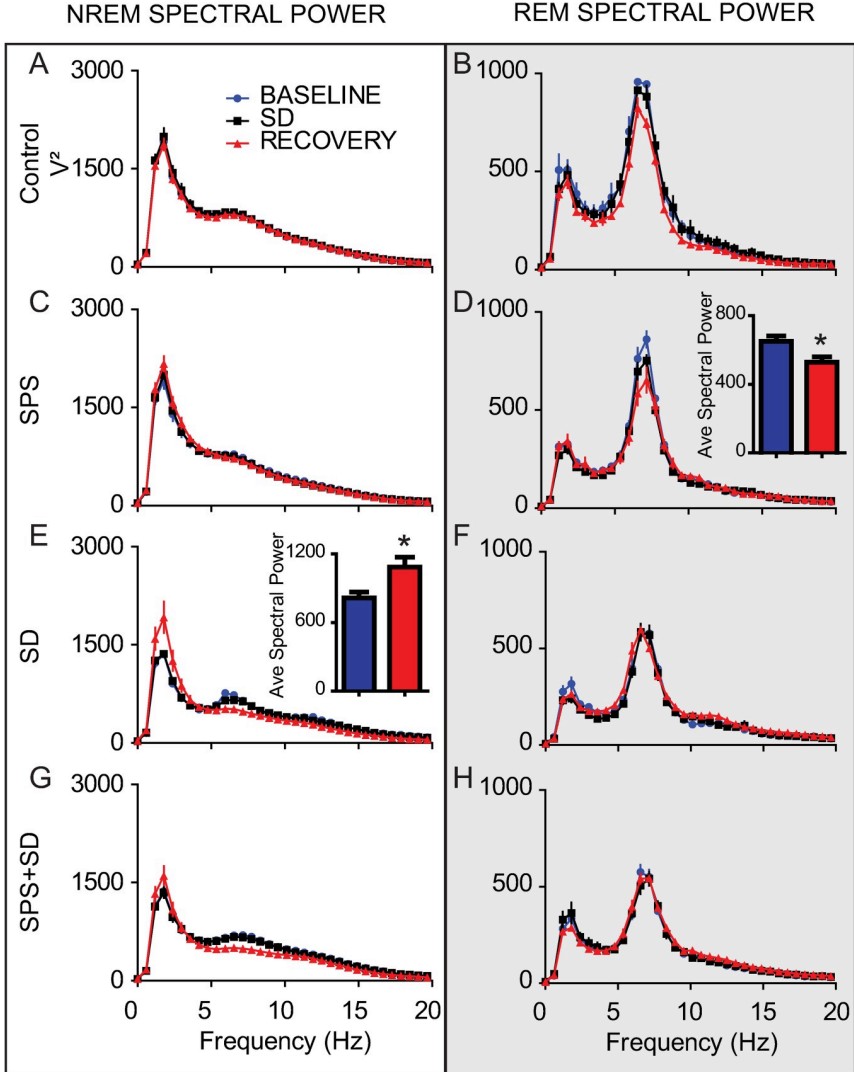

**Fig 5. The combination of SPS and sleep deprivation ameliorates the increase of REM and NREM spectral power induced by either alone.** Each line represents the average spectral power (ZT12-0) on the baseline, sleep deprivation and recovery day over the 0-20Hz frequency range during either NREM sleep (left column) or REM sleep (right column, shaded). Values are shown for control animals (A, B), animals exposed to SPS (C, D), or SD (E, F), or SPS+SD (G, H). Comparisons of differences from baseline over the theta range (4–8 Hz) and delta range (0-4Hz) were preformed using 2-tailed students T-test and are shown in the insets of D and E respectively. Significant differences and are denoted by * (P < 0.05).

## Supporting information

**S1 Data. Compiled raw data for Figs 2–5.**
(XLSX)

## Acknowledgments

We thank Will Clegern, Dominic Brenner, and Daniel Harvey for technical and engineering support in hardware and software components critical to the successful completion of this work. We also thank Michelle Schmidt who assisted in data analysis and the WSU Spokane vivarium and research core facility staff, including Merle Heineke, Mira Deberry, Robert Archuleta, and Megan Chastain.

## Author Contributions

**Conceptualization:** Christopher J. Davis, William M. Vanderheyden.

**Data curation:** Christopher J. Davis, Jason R. Gerstner, William M. Vanderheyden.

**Formal analysis:** Christopher J. Davis, William M. Vanderheyden.

**Funding acquisition:** Christopher J. Davis, William M. Vanderheyden.

**Investigation:** Christopher J. Davis, Jason R. Gerstner, William M. Vanderheyden.

**Methodology:** Christopher J. Davis, William M. Vanderheyden.

**Project administration:** William M. Vanderheyden.

**Resources:** Christopher J. Davis, Jason R. Gerstner, William M. Vanderheyden.

**Software:** William M. Vanderheyden.

**Supervision:** William M. Vanderheyden.

**Validation:** Christopher J. Davis, Jason R. Gerstner, William M. Vanderheyden.

**Visualization:** William M. Vanderheyden.

**Writing – original draft:** William M. Vanderheyden.

**Writing – review & editing:** Christopher J. Davis, Jason R. Gerstner, William M. Vanderheyden.

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
