## [Decision Letter · Decision Letter 0]

29 Oct 2020

PONE-D-20-25539

Single Prolonged Stress Blocks Sleep Homeostasis and Pre-Trauma Sleep Deprivation Does Not Exacerbate the Severity of Trauma-Induced Fear-Associated Memory Impairments

PLOS ONE

Dear Dr. Vanderheyden,

Thank you for submitting your manuscript to PLOS ONE. After careful consideration, we feel that it has merit but does not fully meet PLOS ONE’s publication criteria as it currently stands. Therefore, we invite you to submit a revised version of the manuscript that addresses the points raised during the review process.

We look forward to receiving your revised manuscript.

Kind regards,

Alexandra Kavushansky, PhD

Academic Editor

PLOS ONE

Journal Requirements:

2.Thank you for stating the following in the Funding Source Section of your manuscript:

[This work was supported by the Department of Defense, Congressionally Directed Medical

39 Research Program, Discovery Award # W81XWH-16-1-0319]

 [The funders had no role in study design, data collection and analysis, decision to publish, or preparation of the manuscript.]

Additional Editor Comments (if provided):

In addition to the points raised by the reviewers, I'd like to ask which test was used to verify the normality of data distribution prior to proceeding to further analyses. Was the Bonferroni analysis used as a post-hoc test in every part of the study?

Reviewers' comments:

Reviewer's Responses to Questions

**Comments to the Author**

1. Is the manuscript technically sound, and do the data support the conclusions?

Reviewer #1: Partly

Reviewer #2: Yes

Reviewer #3: Partly

2. Has the statistical analysis been performed appropriately and rigorously? 

Reviewer #1: Yes

Reviewer #2: Yes

Reviewer #3: No

3. Have the authors made all data underlying the findings in their manuscript fully available?

Reviewer #1: Yes

Reviewer #2: Yes

Reviewer #3: Yes

4. Is the manuscript presented in an intelligible fashion and written in standard English?

Reviewer #1: Yes

Reviewer #2: Yes

Reviewer #3: Yes

5. Review Comments to the Author

Reviewer #1: Authors hypothesized that sleep deprivation (SD) prior to trauma exposure may increase the severity of a PTSD-like phenotype in rats exposed to single prolonged stress (SPS), a rodent model of PTSD and investigated "the contribution of pre-trauma sleep loss to subsequent trauma-dependent fear-associated memory impairmen". They also announced that that they identified a unique time frame where trauma exposure and sleep may interact and identifies this window of time as a potential therapeutic treatment window for staving off the negative consequences of trauma exposure. However, the results in this manuscript might not give these conclussion.

1. Because the REMs and NREMs in SD and SD+SPS groups showed no difference, it chould not be conclused that "Treatment-specific sleep phenotypes are shown for the SPS and SPS+SD animals".

2. What are the practical clinical implications of poor sleep prior to trauma for PTSD?

Reviewer #2: • In the entire article, the units should be abbreviated according to the SI system. (e.g.: 15 minutes: 15 min, 2 hours: 2h, 21-24 degree C: 21-24 �C).

• It was reported that male rats were used in the study. The reason for using especially male rats in the experiments should be stated in the animal's section.

• Adding an experimental design drawing that schematizes the experimental protocols used by the researchers makes the article both easier to understand and more visual.

• It would be more appropriate to write 72.99 ±7.9 % instead of M = 72.99 ±7.9 percent in statistical expressions.

Reviewer #3: In the present manuscript, the authors investigated the effects of sleep deprivation on PTSD-like symptoms following trauma. The authors hypothesized that sleep deprivation (SD) prior to trauma exposure may increase the severity of a PTSD-like phenotype in rats exposed to single prolonged stress (SPS). However, they found that sleep loss prior to trauma exposure did not further exaggerate the PTSD-like

phenotype as there was no difference in freezing between the SPS and the SPS+SD group on the fear recall day. The findings are interesting, but the manuscript have major fundamental problems as mentioned below:

1. The authors claim that fear-associated memory processing was impaired after SPS was incorrect since there was an increase in the freezing percent in SPS and SPS-SD groups during the extinction phase which suggest that animals retained their memory about the trauma associated with the cue.

2. Another problem with the results was that they did not find any effect of extinction in the SPS group. On the fear recall day, animals in the SPS group displayed significant increase in the freezing.

3. In the SD group, they did not find any increased freezing during fear conditioning which they did not discuss in the discussion. But interestingly, their freezing percent increased during fear extinction. Again, no explanation in the discussion.

4. In addition, there was difference in the degree of freedom in Figure 1. In fear condition, it was 19, in fear extinction, it was 119 whereas in fear recall, it was 39. I do not understand why authors have used different number of animals on each of these three days.

6. PLOS authors have the option to publish the peer review history of their article (what does this mean?). If published, this will include your full peer review and any attached files.

Reviewer #1: No

Reviewer #2: **Yes: **Asli Aykac

Reviewer #3: No

---

## [Author Response · Author response to Decision Letter 0]

5 Nov 2020

The full response to reviewers has been submitted here as a word document.

---

## [Decision Letter · Decision Letter 1]

27 Nov 2020

Single Prolonged Stress Blocks Sleep Homeostasis and Pre-Trauma Sleep Deprivation Does Not Exacerbate the Severity of Trauma-Induced Fear-Associated Memory Impairments

PONE-D-20-25539R1

Dear Dr. Vanderheyden,

We’re pleased to inform you that your manuscript has been judged scientifically suitable for publication and will be formally accepted for publication once it meets all outstanding technical requirements.

Kind regards,

Alexandra Kavushansky, PhD

Academic Editor

PLOS ONE

Additional Editor Comments (optional):

Reviewers' comments:

Reviewer's Responses to Questions

**Comments to the Author**

1. If the authors have adequately addressed your comments raised in a previous round of review and you feel that this manuscript is now acceptable for publication, you may indicate that here to bypass the “Comments to the Author” section, enter your conflict of interest statement in the “Confidential to Editor” section, and submit your "Accept" recommendation.

Reviewer #2: All comments have been addressed

Reviewer #3: All comments have been addressed

2. Is the manuscript technically sound, and do the data support the conclusions?

Reviewer #2: Partly

Reviewer #3: Yes

3. Has the statistical analysis been performed appropriately and rigorously? 

Reviewer #2: Yes

Reviewer #3: Yes

4. Have the authors made all data underlying the findings in their manuscript fully available?

Reviewer #2: Yes

Reviewer #3: Yes

5. Is the manuscript presented in an intelligible fashion and written in standard English?

Reviewer #2: Yes

Reviewer #3: Yes

6. Review Comments to the Author

Reviewer #2: (No Response)

Reviewer #3: (No Response)

7. PLOS authors have the option to publish the peer review history of their article (what does this mean?). If published, this will include your full peer review and any attached files.

Reviewer #2: **Yes: **Asli Aykac

Reviewer #3: No

---

## [Editor Report · Acceptance letter]

8 Dec 2020

PONE-D-20-25539R1 

Single prolonged stress blocks sleep homeostasis and pre-trauma sleep deprivation does not exacerbate the severity of trauma-induced fear-associated memory impairments 

Dear Dr. Vanderheyden:

I'm pleased to inform you that your manuscript has been deemed suitable for publication in PLOS ONE. Congratulations! Your manuscript is now with our production department. 

Kind regards, 

on behalf of

Dr. Alexandra Kavushansky 

Academic Editor

PLOS ONE